# Tailoring van der Waals dispersion interactions with external electric charges

Andrii Kleshchonok[1] & Alexandre Tkatchenko[2]

van der Waals (vdW) dispersion interactions strongly impact the properties of molecules and materials. Often, the description of vdW interactions should account for the coupling with pervasive electric fields, stemming from membranes, ionic channels, liquids, or nearby charged functional groups. However, this quantum-mechanical effect has been omitted in atomistic simulations, even in widely employed electronic-structure methods. Here, we develop a model and study the effects of an external charge on long-range vdW correlations. We show that a positive external charge stabilizes dispersion interactions, whereas a negative charge has an opposite effect. Our analytical results are benchmarked on a series of (bio)molecular dimers and supported by calculations with high-level correlated quantum-chemical methods, which estimate the induced dispersion to reach up to 35% of inter-molecular binding energy ($4\,kT$ for amino-acid dimers at room temperature). Our analysis bridges electrostatic and electrodynamic descriptions of intermolecular interactions and may have implications for non-covalent reactions, exfoliation, dissolution, and permeation through biological membranes.

[1] Fritz-Haber-Institut der Max-Planck-Gesellschaft, Faradayweg 4-6, 14195 Berlin, Germany. [2] Physics and Materials Science Research Unit, University of Luxembourg, Luxembourg City L-1511, Luxembourg. Correspondence and requests for materials should be addressed to A.T. (email: alexandre.tkatchenko@uni.lu)

**V**an der Waals (vdW) dispersion interactions play an important role in the structure formation, energetic stability, and reaction mechanisms for a large variety of molecules and materials[1–4]. These interactions originate from spontaneous charge oscillations that consequently induce fluctuating multipole moments on surrounding atoms and molecules. As vdW interactions do not require the presence of permanent multipole moments, they arise between any polarizable bodies. Even though these intermolecular interactions are typically much weaker than intramolecular ones, they are responsible for many observable macroscopic phenomena[5].

In addition to internal permanent and fluctuating electrostatic moments, molecules are substantially affected by external electric fields of various origins. For instance, in biological systems, intermolecular interactions usually occur in solvents with certain salt concentration or in acidic environments[1,6–8]. One of the most illustrative examples are biological membranes, formed by lipid bilayers, which constitute an essential component of a living cell. Dispersion energy contributes to the interlayer interaction and its properties can be modified by changing the salt concentration[9,10].

The synthesis and tuning of materials properties is another area where external fields serve as an essential tool. For example, graphene can be exfoliated from graphite in the presence of a weak electric field[11,12], and subsequently used in the synthesis of nanoribbons and carbon nanotubes[13,14], in catalysis[15,16], and in dissociation reactions[17]. In addition, the observation of reaction acceleration in positively charged microdroplets[18,19] demonstrates that reaction rates depend on the polarity of the droplet. Charged droplets are also employed in the soft-landing technique to form non-covalent molecular nanostructures on surfaces under ambient conditions[20,21].

These examples taken together strongly suggest that intermolecular interactions can be tailored by electric fields. Obviously, electrostatic contributions arising from static external fields are already included in electronic-structure calculations. However, in many systems the dominant intermolecular interactions are of electrodynamic origin. Naively, one would expect that only time-dependent oscillating electric fields should have a significant impact on spontaneous charge oscillations and, therefore, on van der Waals dispersion interactions. However, here we show that an external static charge can also substantially affect intermolecular dispersion interactions. The correlation energy term derived herein is not included either in standard or dispersion-corrected density-functional theory (DFT), semi-empirical and second-order quantum perturbation theory (MP2), or classical force fields

(see for example binding energies obtained by different numerical methods in Table 1). Hereafter, we will refer to the vdW part of intermolecular correlation energy that is modified by an external charge as field-induced dispersion (FID).

The calculation of FID energies requires the development of quantum-mechanical techniques beyond second-order perturbation theory and simultaneously consider multipole expansion terms beyond the dipole approximation in the treatment of electronic correlation. This allows to account for the effect of an electric field on excited states, while conventional DFT + vdW approaches capture influence of electric field on static density only. Therefore, FID is included only in excessively expensive electronic-structure approaches, such as the random-phase approximation (RPA), coupled cluster, or quantum Monte Carlo methods. In this work, we generalize the treatment of vdW interactions to include the effect of an inhomogeneous electric field and demonstrate an analytical method of evaluating the FID contribution to binding energies, thus bridging electrostatic and electrodynamic models of intermolecular interactions. This paves the way to efficiently include FID energies in classical force fields and electronic-structure calculations.

## Results

**Intermolecular interactions from quantum Drude oscillators.** Electrons in systems with finite band gaps, such as organic molecules, nonmetallic solids, and nanostructures, are well described by a localized representation. Therefore, collective charge density fluctuations in these systems arise from the dynamically correlated motions of local multipole excitations. Accordingly, we treat the response of valence electrons of a given molecule as that of a set of interacting atomic response functions. Quantum-mechanical parameterization of the valence electronic response in terms of coupled atomic fluctuations is done efficiently and accurately within the quantum Drude oscillator (QDO) approach[22–31]. The QDO model replaces oscillations of the electron cloud on each atom with an effective quantum harmonic oscillator, characterized by a set of three effective parameters: mass, frequency, and charge ($\mu$, $\omega$, $q$)[24–26].

The coupled Drude oscillator model has been shown to yield a quantitatively accurate description of many-body dispersion interactions in the dipole limit[29,31–33], which makes it a promising approach for higher multipole generalization and coupling to external electric fields[26,28,34,35].

The Hamiltonian of a system of $N$ QDOs interacting via Coulomb forces between themselves and $M$ point charges $\delta_j$,

---

**Table 1 Field-induced dispersion (FID) energy, defined by Eq. (2), and its comparison with ab initio RPA and MP2 methods**

| Molecular dimer | $E_{FID}$, meV (analytic) | $W_c^{bind}(\delta)$, meV (RPA@PBE0) | $W_c^{bind}(\delta)$, meV (MBD@PBE0) | $\Delta E_c^{bind}(+\delta)$, meV (MP2@HF) | $\Delta E_c^{bind}(-\delta)$, meV (MP2@HF) | $E_{total}^{bind}(\delta=0)$, meV (RPA@PBE0) |
|---|---|---|---|---|---|---|
| Ammonia | 26.2 | 20.0 | 0.3 | −14.6 | 7.2 | 134.2 |
| Cyclopentane | 47.4 | 62.8 | 0.9 | −43.3 | 21.7 | 134.6 |
| Neopentane | 11.5 | 13.0 | 0.6 | −8.1 | 12.9 | 81.4 |
| Ethyne | 6.3 | 6.0 | 0.5 | −9.0 | 3.5 | 69.5 |
| Benzene* | 11.5 | 12.8 | 0.3 | −13.8 | 9.9 | 96.5 |
| N-methylacetamide* | 21.0 | 17.9 | 3.2 | −7.2 | 3.9 | 339.9 |
| Pyrazine* | 17.0 | 9.6 | 0.7 | −11.3 | 7.6 | 179.2 |
| Pyridine* | 9.8 | 9.0 | 0.6 | −9.5 | 6.9 | 156.5 |
| Water | 0.6 | 1.8 | 0.0 | 0.0 | 0.0 | 150.9 |

First column: analytical $E_{FID}$ energy, which can be directly compared with the reference RPA correlation contribution to the binding energies $W_c^{bind}(\delta) = \frac{1}{2}[E_c^{bind}(-\delta) - E_c^{bind}(\delta)]$, shown in the second column; third column: $W_c^{bind}(\delta)$ for the MBD method, evaluating conventional dipole–dipole vdW interactions; fourth and fifth columns: the effect of an external charge on MP2 correlation binding energies. In the last column, total RPA-binding energies without an external field are provided to put the FID energy magnitude into perspective. With the "*" symbol we denote cases with 5 Å distance from the dimer center of mass to the external charge, or 3 Å otherwise

placed at $\widetilde{R}_j$, is given by:

$$
H = \sum_i^N H_{0i} + \sum_i^N \sum_j^M \delta_j q_i \left( \frac{1}{|R_i - \widetilde{R}_j|} - \frac{1}{|r_i - \widetilde{R}_j|} \right)
$$
$$
+ \frac{1}{2} \sum_{i \neq i'}^N q_i q_{i'} \left( \frac{1}{|R_i - R_{i'}|} + \frac{1}{|r_i - r_{i'}|} \right. \tag{1}
$$
$$
\left. - \frac{1}{|r_i - R_{i'}|} - \frac{1}{|R_i - r_{i'}|} \right),
$$

where $H_{0i} = -\frac{\hbar^2}{2\mu_i}\nabla_{r_i}^2 + \frac{1}{2}\mu_i \omega_i^2 (R_i - r_i)^2$ is the unperturbed QDO Hamiltonian, assuming fixed oscillation center position $R_i$ and $r_i$ is a position of the Drude particle. Details of the unperturbed QDO solution may be found in the Supplementary Note 1. Equation (1) is similar to the conventional molecular Hamiltonian, however the full electronic-nuclear system is replaced by Drude quasiparticles, placed on each atom. Owing to its quantum nature, the QDO represents a spatial distribution of charge, which could be modified by other QDOs or external fields, giving this model the ability to capture complex electronic response effects. Owing to the full Coulomb coupling between charges, the exact quantum-mechanical solution of the QDO Hamiltonian in Eq. (1) contains all multipoles and many-body effects to all orders of perturbation theory[26]. This makes the QDO Hamiltonian an optimal starting point to develop approximations. For example, within classical mechanics one would recover polarizable force fields[36–38], whereas within the quantum-mechanical dipole approximation one obtains the previously developed many-body dispersion (MBD) model[29].

Here, we study the effect of an electric field induced by an external point charge on dispersion interactions between QDOs in the linear response regime. We initially consider a system made of two QDOs, $A$ and $B$, and a point charge $\delta$, placed at distances $\widetilde{R}_{A/B}$ from them (see Fig. 1). The quantum-mechanical Drude model allows to describe electrostatics, induction, and dispersion effects in isolation and induced by external charges. The approach presented here could be generalized to a set of external charges and QDOs in a straightforward manner.

The last two terms in Eq. (1) could be treated as perturbation $H' = H_A + H_B + H_{AB}$ of $H_0$, which are caused by the interaction of each QDO with an external point charge $H_{A/B}$, and the interaction between QDOs, $H_{AB}$. Henceforth, we use the indices $A$ and $B$ to mark variables related to different oscillators. In terms of the spherical multipole tensor $Q_{lm}$ (of order $l$, $-l \leq m \leq l$), the interaction of test charge $\delta$ with a negatively charged Drude quasiparticle and positively charged oscillation center is expressed in a compact form[3,26]: $H_{A/B} = -\delta \sum_{l=1}^{\infty} \sum_{m=-l}^{m=l} Q_{lm}^{A/B} / \widetilde{R}_{A/B}^{l+1}$.

The interaction between two QDOs at a distance $R$ between them is given by the multipole interaction tensor $T_{lm;l'm'}^{AB}$, which depends on this distance and relative orientation[3,26,39–43]: $H_{AB} = \sum_{l,l'=0}^{\infty} \sum_{m,m'} Q_{lm}^A T_{lm;l'm'}^{AB} Q_{l'm'}^B$. The response to an external electric field produced by a charge is given by the polarizabilities $\alpha_{lm,l'm'}$, defined in second-order Rayleigh-Schrödinger perturbation theory[3,26]. Herein, we consider input atomic polarizabilities to be isotropic $\alpha_{lm,l'm'} = \alpha_l \delta_{l,l'} \delta_{m,m'}$, where $l$ defines the multipole order ($l = 1$ corresponds to the dipole, $l = 2$ to quadrupole, etc.) and $\alpha_l$ is given by the analytical expression $\alpha_l = \left(\frac{q^2}{\mu\omega^2}\right)\left[\frac{(2l-1)!!}{l}\right]\left(\frac{\hbar}{2\mu\omega}\right)^{l-1}$[26]. This is an excellent approximation within the employed atomic Tkatchenko–Scheffler model (see below and in ref. [44]).

**Analytic expression for the field–induced dispersion energy.** Building the perturbation theory on eigenvectors of $H_0$ and treating $H'$ as a perturbation, Martyna and co-authors developed a Jastrow-type diagrammatic technique[26,45], which proves to be a very useful tool for understanding coupled QDOs. In brief, diagrammatic rules are the following: the yellow blocks are associated with the electrostatic multipole interaction between two QDOs and their number indicates successive orders of perturbation expansion; connectors, coming out of one end of a yellow bar, show the multipole order. The asymptotic interaction power laws are given by combination of $R$ and $\widetilde{R}$ and can be obtained by analyzing separate parts of diagrams: yellow bar gives $R'^{-1}$ decay and connector contributes as $R'^{-2}$, where $R' \in \{R, \widetilde{R}\}$. For the detailed discussion of the diagrammatic principles we refer the reader to ref. [26] and references therein. Several representative diagrams for two (three) QDOs are shown in Fig. 2. Within this framework, dispersion terms arise from the bubble-like closed diagrams (Fig. 2c, f, h, i) and the open ends indicate the polarization terms (Fig. 2b, d, e), which arise from interaction with external fields.

In this work, we focus on the leading dispersion term modified by an external charge $\delta$, corresponding to the dipole–quadrupole polarization–dispersion interaction (Fig. 2i), which appears in the third order of perturbation theory. This diagram arises from the dispersion interaction between two QDOs, one being in a dipolar state, the other in an excited quadrupolar state owing to polarization via the external charge. The diagram on Fig. 2i is not symmetric, as it relates to the polarization of one out of two interacting QDOs. However, the total dipole–quadrupole charge-induced dispersion (FID–(DQ)) energy should be completed with another diagram that relates to the quadrupole polarization of the second QDO. The sum of these two diagrams will be referred to as $E_{\mathrm{FID}}$ in this study.

This first non-trivial FID diagram, shown on Fig. 2i, translates into the analytic form as follows (details of the derivation and general form of the dispersion terms are provided in Supplementary Note 1 along with discussion of Fig. 2g):

$$
E_{\mathrm{FID}}^A = -\frac{\delta}{2} \frac{\alpha_1^B \alpha_2^A \omega_A \omega_B}{(2\omega_A + \omega_B)(\omega_A + \omega_B)} \frac{1}{\widetilde{R}_A^2}
$$
$$
\times \sum_{m_A, m_B} T_{2,-m_A;1,-m_B} T_{1,m_A;1,m_B} \sqrt{4 - m_A^2} \tag{2}
$$

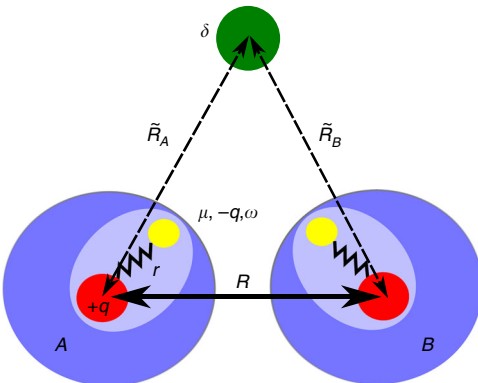

**Fig. 1** Model for field-induced dispersion. A system made of two QDOs with distance $R$ between them and a point charge $\delta$ placed at a distance $\widetilde{R}_A$ and $\widetilde{R}_B$ from the oscillator centers $A$ and $B$, respectively. QDOs are sketched by blue circles with red oscillation center and yellow oscillating Drude quasiparticle. The external charge is marked by a green dot

Equation (2) depends quadratically on $\widetilde{R}_A^{-1}$ and linearly on $\delta$, reflecting that only one QDO is polarized by the external charge. This dispersion term scales as $R^{-7}$ with distance between two oscillators, which makes it noticeable compared with the

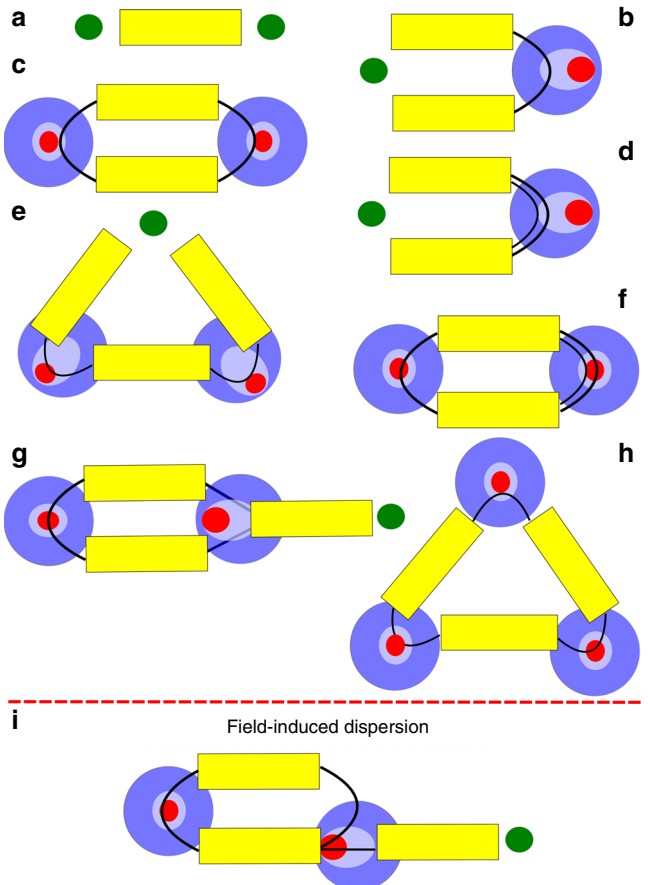

**Fig. 2** Diagrammatic expansion of interacting quantum Drude oscillators. Representative low-order interaction energy diagrams for coupled QDOs, associated with the following expansion terms: **a** bare Coulomb electrostatic interaction; **b** dipole polarization; **c** conventional vdW dipole–dipole dispersion; **d** quadrupole polarization; **e** many-body dipole polarization; **f** dipole–quadrupole dispersion; **g** dipole–quadrupole electrostatic interaction; **h** three-body dipole–dipole–dipole dispersion; **i** charge-induced pairwise dipole–quadrupole dispersion (called FID in this paper). QDOs are sketched by blue circles with red oscillation center and the external charge is shown with a green dot, polarization is indicated with elongation of the inner circle

conventional dipole–dipole dispersion interaction decaying as $R^{-6}$. Also we note that the FID energy is a purely quantum effect, as $\alpha_2 \propto \hbar$ vanishes in the classical limit for a QDO[26]. A similar effect is expected in case of other non-uniform electric fields, for example that induced by a finite-size dipole. FID would scale as $\tilde{R}^{-9}$ with the distance to the dipole, making dispersion induced by a point charge a leading-order effect.

In this work, we model each atom by a separate QDO and parametrize it with a Tkatchenko–Scheffler (TS)-like approach[46], extended to the quadrupole polarizability in the following way (see Supplementary Note 1 for details): $\alpha_2^{\mathrm{eff}} = \alpha_2^{\mathrm{free}} \left( \frac{V_h^{\mathrm{eff}}}{V_h^{\mathrm{free}}} \right)^2$. This approach takes into account local electronic environment by weighting the free-atom quadrupole polarizability $\alpha_2^{\mathrm{free}}$[47,48] with the ratio between free and effective Hirshfeld volumes. The angular dependence of dipole–quadrupole FID is expressed by the sum over angular momentum indices $m$ of the interaction function product. It takes the simplest form when two identical QDOs $\alpha_l^A = \alpha_l^B = \alpha_l$ lie on the same $z$ axis with the same distance

to the external charge $\tilde{R}_A = \tilde{R}_B = \tilde{R}$. In this case, the frequency dependence is canceled out and the sum of two QDOs FID contribution $E_{\mathrm{FID}}^A + E_{\mathrm{FID}}^B$, given by Eq. (2), simplifies to:

$$E_{\mathrm{FID}} = -3\delta\alpha_2\alpha_1 \frac{1}{\tilde{R}^2} \frac{1}{R^7}. \qquad (3)$$

We note that a positive sign of the external charge $\delta > 0$ corresponds to a negative energy contribution $E_{\mathrm{FID}} < 0$. An intuitive mechanism behind this charge dependence is the following: as positive $\delta$ attracts the interacting pair of Drude particles, it polarizes both their orbitals in the same direction toward the external charge, which leads to stabilization. Similar arguments are valid for the inverse effect in the field of a negative charge. These findings could be explained in terms of the Hellmann–Feynman theorem as well. External charge populates excited states of the QDO system, leading to anisotropic delocalization of the electron cloud, which gives rise to Feynman forces[49]. Therefore, alternatively one can derive the FID contribution in terms of high-order hyperpolarizabilities, similarly to the Feynman dipole approach described in ref.[50]. We note, however, that the FID contribution is present already in the linear response regime.

Along with the dispersion term, described by Eq. (2), the total energy of the system contains classical electrostatic interaction terms, which might be important for polar molecules. However, these terms do not scale with molecular polarizability, hence dispersion interactions remain dominant for large molecules. Furthermore, electrostatic interactions are already included in force fields and DFT calculations employing semilocal functionals. In contrast, to the best of our knowledge FID terms have so far not been included in any atomistic simulation. Next, we will show that FID contributes significantly to the binding energies of relatively small molecules and its importance grows with molecular size and polarizability.

**Field–induced dispersion interaction between small molecular dimers.** Before presenting the importance of FID effects in biological systems, we first discuss the FID contribution to the correlation part of the binding energy between small molecular dimers and benchmark our analytic results (see Eq. (2)) with direct ab initio calculations. Ab initio calculations were carried out using the RPA method[51,52] converged at a complete one-electron basis set limit (see Methods section for details), and these results serve as an accurate reference for comparison with analytic FID values. The RPA approach computes the correlation energy to infinite order in perturbation theory and treats the influence of an external charge on excited states of molecules, hence being an appropriate reference for FID.

We start by benchmarking the FID effect on a molecular cyclopentane dimer system. Figure 3 shows $E_{\mathrm{FID}}$ as a function of the cyclopentane dimer center of mass separation, whereas keeping fixed the distance from each QDO to the external charge, $\tilde{R}_A = \tilde{R}_B = 5$ Å. The interaction energy computed with the analytic formula in Eq. (2) is in excellent agreement with reference RPA calculations. In addition, both RPA and analytic FID formula are essentially antisymmetric with respect to the sign of the external charge. In contrast, all popular methods for vdW interactions in DFT are unable to capture the FID effect, as the direct change in the electron density by external charge is negligible (identical conclusions hold for vdW methods such as D3, vdW-DF, XDM, TS, and MBD[29,46,53–57]). Molecular orbitals employed in second-order Møller–Plesset perturbation theory (MP2) framework are more responsive to electric fields than the ground-state electron density. Hence, MP2 correlation energy is affected by external charges.

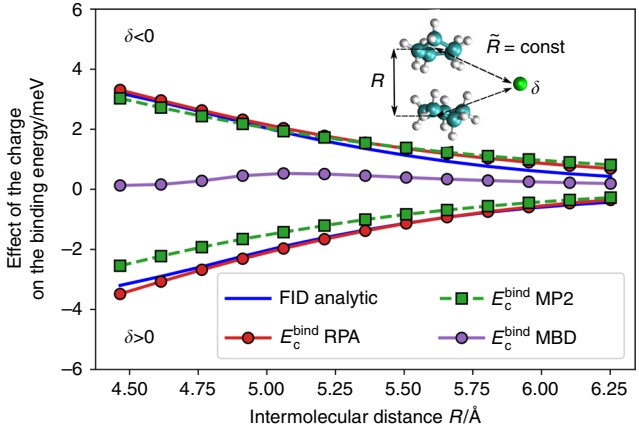

**Fig. 3** Field-induced dispersion effect in cyclopentane dimer. FID as a function of intermolecular distance $R$ in Å, calculated at a fixed distance to the external unit charge $\tilde{R} = 5$ Å for the cyclopentane dimer (geometry shown as inset). The analytical result is shown in blue. The reference RPA@PBE0 calculations are marked with a red curve. The effect of the charge in conventional vdW methods such as MBD@PBE0 (purple) is negligible. The effect of the charge on MP2@HF correlation energies (dashed green) is visible, however having a different origin than FID. Upper positive halfplane corresponds to the negative sign of the external charge $\delta = -1$, and negative—to the positive sign $\delta = 1$

However, MP2-binding energies in this case arise from hyperpolarization effects (not linear polarization as in RPA) and are not antisymmetric with respect to $\delta \to -\delta$, having a smaller magnitude compared to the reference RPA calculations (also see Table 1). This is particularly significant for practically relevant distances, when the external charge is close to the molecule.

Extending our study to larger systems, we tested the magnitude of FID effect on molecular dimers from the S66 benchmark database of non-covalent interactions[58] (see Table 1). We consider dimer geometries at equilibrium separation[58]. We chose the position of the external charge such that the electric field at the center of mass of the dimer reaches $10^9$–$10^{10}$ V/m, by placing an external charge at 3 or 5 Å. This range of distances reflects realistic environments present in biological systems. FID calculations are summarized in Table 1, where the following notation is used. The molecular dimer correlation binding energy in the presence of an external charge $\delta$ is defined as: $E_c^{\text{bind}}(\delta) = E_c^{AB}(\delta) - E_c^A(\delta) - E_c^B(\delta)$, where $E_c^{AB}$ is a correlation energy of the molecular dimer, and $E_c^{A/B}$ are the correlation energies of the separate moieties $A$ and $B$. The effect of the external charge on correlation energy is given by the difference: $\Delta E_c^{\text{bind}}(\delta) = E_c^{\text{bind}}(\delta) - E_c^{\text{bind}}(\delta = 0)$. In order to separate the vdW dispersion contribution in Eq. (2) from other terms of electrostatic-induction origin (see Supplementary Fig. 1) we use the fact that FID changes sign with $\delta$ and define the following linear combination: $W_c^{\text{bind}}(\delta) = \frac{1}{2}\left[E_c^{\text{bind}}(-\delta) - E_c^{\text{bind}}(\delta)\right]$. Overall, our analytic formula for FID yields an excellent agreement with full RPA calculations. Largest deviations amount to 30% and stem mainly from the fact that our formula is the leading-order term, whereas RPA includes contributions up to infinite order. Our model can be generalized to infinite order in a straightforward way as a potential future work.

There are several noteworthy observations that can be drawn from Table 1. First, the FID energy can reach up to 35% (47.4 meV) of the total binding energy for the cyclopentane dimer. FID also contributes substantially in benzene, N-methylacetamide and pyrazine dimers. Second, it is clear that existing methods for vdW dispersion energy in DFT[29,46,53–57] are unable to describe FID because they do not have any mechanism to couple to external electric fields. In contrast, MP2

correlation energies are affected by the external charge due to orbital polarization via the external charge. However this effect is smaller than reference RPA@PBE0 results and non-symmetric with respect to the change of the external charge sign (compare second column in Table 1 with fourth and fifth columns).

For non-polar molecules (i.e., ammonia, cyclopentane, N-methylacetamide), FID energies are larger or comparable to the electrostatic contribution to the binding energy in the presence of an external charge (see Supplementary Table 1). In general, electrostatics will be dominant for polar molecules (i.e., water dimer in the last row of Table 1). However, the sign of the FID contribution is proportional to $-\text{sign}(\delta)$ in contrast to the charge-induced electrostatic term, which depends on the relative orientation of the molecular dipole. Consequently, in many practical situations, we expect an interplay between electrostatic energies and FID and tuning this delicate balance paves a novel way to control structure and dynamics via externally applied electric fields.

**Field-induced dispersion interaction in biological systems.** In this section, we discuss the importance of FID in biological systems, namely in ionic channel models (Fig. 4) and amino-acid dimers (Fig. 5), both of which are key to biomolecular function. Amino acids serve as building blocks for proteins, which are in turn essential components of living cells. Amino acids are responsible for ion channel formation that serve for selective ion transport through the cell membrane[7]. Ion mobility can be controlled by external stimuli such as voltage, mechanical stress, protein phosphorylation, ligand binding, or changes in pH[7]. The last two gating mechanisms occur in the presence of external charges, and here we show that FID energies may contribute substantially to these processes.

We start with a set of amino-acid dimers at equilibrium separation with their geometries optimized using the DFT + MBD[29] method in the absence of external fields. Subsequently, a point charge was added at 3.5 Å distance from the center of mass of the dimer and the FID energy was determined, by carrying out RPA calculations (see Supplementary Table 2 for details). The results are summarized in Fig. 5, where we show FID contributions obtained from RPA correlations and the analytical formula in Eq. (2). In order to assess the significance of the FID contribution, we compare it with the dispersion binding energy obtained with MBD for the amino-acid dimer in absence of any field. Remarkably, FID contribution reaches up to 35% of the dipole–dipole dispersion binding energy, varying in the range from 3 to 15% of the total binding energy. Hence, we conclude that FID energies for biological systems can substantially exceed 1 kcal/mol—the minimum desired level of accuracy for atomistic modeling.

As a final example we consider the selectivity filter region of the Kcv ion channel[59] and dispersion interactions therein. We focus on one of the preferred binding sites—S4 (shown in Fig. 4a), that is typically composed of four threonine residues that provide four backbone carbonyl oxygens and four side-chain hydroxyl oxygens for ion coordination[59–61]. S4 site plays a special role in the ion gating mechanism, as $Ba^{2+}$ ions can bind to this site as well, blocking $K^+$ permeation[60,62]. Besides ion substitution, sequence decoding shows that within the channel a threonine residue could be replaced by serine (identical to threonine except for one missing methyl group)[60] (see Fig. 4a). Site-directed mutagenesis experiments suggest that threonine-to-serine (T → S) substitution in the S4 sites reduces the channel susceptibility to $Ba^{2+}$ and its overall opening probability[61]. The key role of methyl groups for understanding the polarization and as a consequence vdW interactions in Kcv channels as well as dramatic ion affinities reduction during T → S substitutions was already suggested[61]. This leads us to the question of whether an inhomogeneous

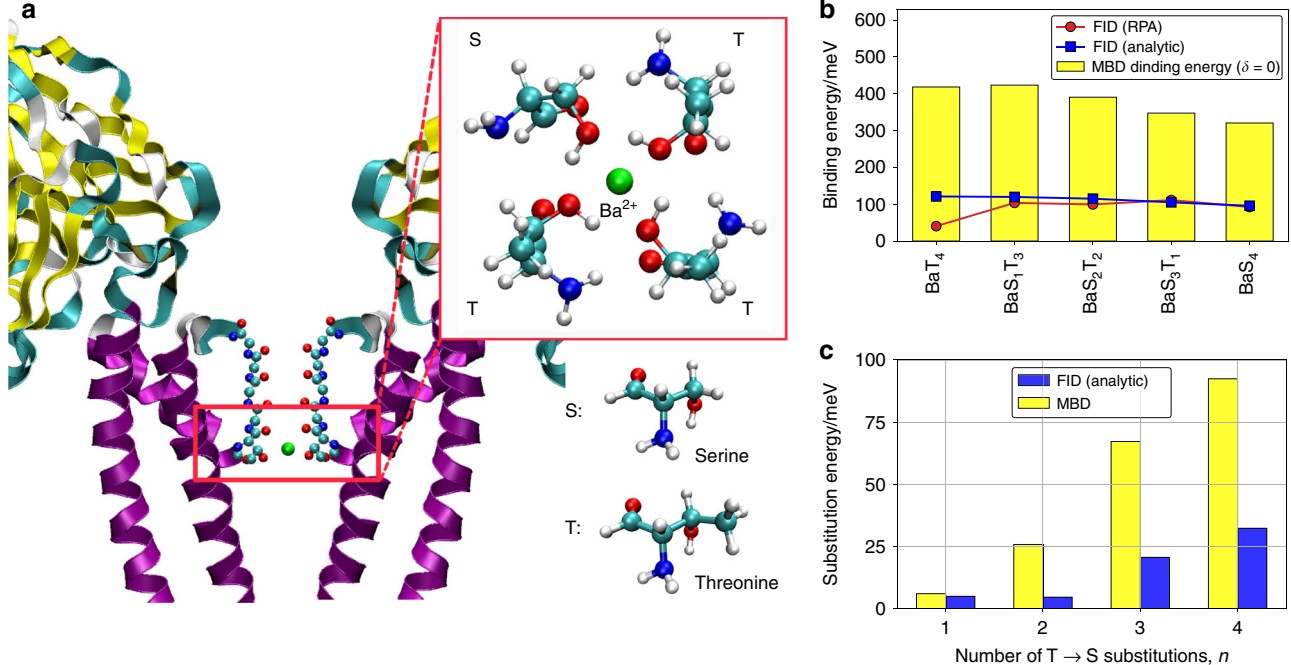

**Fig. 4** Field-induced dispersion effect in the selectivity filter of Kcv ion channel. **a** Cross section of selectivity filter of a representative $Ba^{2+}$ ion channel[59] and zoomed S4 region with three threonine and one serine molecule ($BaT_3S_1$); **b** Absolute value of FID $W_c^{bind}(\delta = +2)$ (in meV) obtained from numerical RPA correlations (red line) and analytic Eq. (2) (blue line) compared with the MBD energy (yellow bars) for the $Ba^{2+}$ complex with four amino-acid ligands; **c** Substitutional MBD energies (shown in yellow) and substitutional FID energy contribution (blue) for the same ion complexes

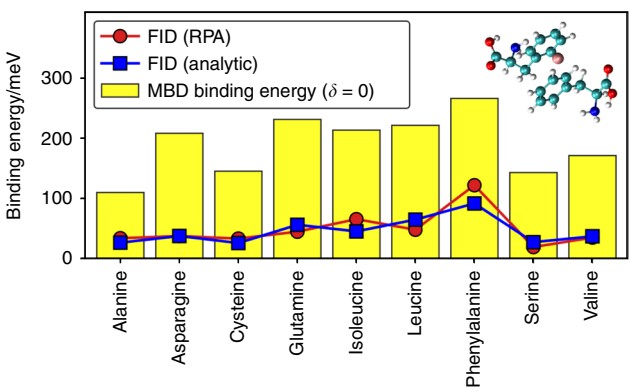

**Fig. 5** Field-induced dispersion in amino-acid dimers. FID $W_c^{bind}(\delta = 1)$ energy (in meV) obtained from RPA calculations (red line) and analytic Eq. (2) (blue line) compared with the MBD binding energy (yellow bars). The geometry of phenylalanine dimer and a point charge (in pink) is shown in the inset

electric field could have considerable effect on dispersion interactions and binding in Kcv channel beyond direct electron density polarization. To address this question, we first compare FID energies to the MBD-binding energy of $Ba^{2+}$ (see Supplementary Fig. 2 for $K^+$ and $Na^+$ ions) for isolated S4 site geometries, composed of threonine or serine residues (Fig. 4b). As the ions are positively charged, FID stabilizes dispersion energy with a relative contribution ranging from 10% in the case of $K^+$ to 30% for $Ba^{2+}$ complex, having a considerable effect in the total binding energy (total energies are shown in the Supplementary Tables 3–5).

To better understand the role of FID and its contribution to the vdW dispersion energy, we now consider substitution energies for the methylation process in ionic channels. The substitution energy is calculated as the energy difference between the left and

right hand side of the following reaction:

$$AT_4 + nS \rightleftharpoons AT_{4-n}S_n + nT, \qquad (4)$$

where $n$ is the unit increment of threonine vs. serine substitutions, $A$ represents one of the $K^+$, $Na^+$, $Ba^{2+}$ ions and $nT$ or $nS$ are amino acids in the gas phase. These substitutions result in minor geometry changes, so we used geometries reported in ref.[61], modeling an ion by a point charge. Corresponding substitution binding energies of the $Ba^{2+}$ complex, where threonine groups were systematically substituted by serine groups, are shown in Fig. 4c (see Supplementary Tables 3–5 for total binding energies). Substitutional energies are also largely affected by FID. One $T \rightarrow S$ substitution ($n = 1$) almost doubles FID stabilization of the dispersion energy and reaches $-35$ meV in the case of complete $T \rightarrow S$ substitution. The main finding here is that FID enhances the methylation stabilization effect. Overall, our results provide compelling evidence that FID is an essential energy contribution to biological systems in the presence of external electric fields.

Figure 4b also indicates the need for further theoretical developments of the FID model. Namely, the largest deviation between reference RPA calculations and analytical FID model occurs for the most polarizable $BaT_4$ system and suggests that higher-order polarization–dispersion terms beyond that shown in Fig. 2i must be included to achieve a quantitative treatment of FID effects.

## Discussion
Our analysis demonstrates the possibility of tailoring intermolecular van der Waals dispersion interactions with external electric charges. Further analysis on graphene exfoliation from graphite (K.A. & T.A., manuscript in preparation) indicates that similar tunability will hold for more general homogeneous and inhomogeneous electric fields. As charged groups and electric fields are ubiquitous in (bio)molecular systems and materials, we expect our findings to have broad implications for modeling and understanding intermolecular interactions in the presence of such

complex environments. For example, FID interactions might substantially affect a range of phenomena, including non-covalent reactions, exfoliation, dissolution, and permeation through biological membranes.

A distinctive aspect of FID interactions is that they can stabilize or destabilize a given complex depending on the sign of the electric charge. Hence, given the fact that FID can reach up to 4 $kT \approx 103$ meV in amino-acid dimers at room temperature, one would expect non-trivial FID effects on molecular orientations, equilibrium distances, and energy ranking of competing polymorphic structures. Furthermore, the dependence of FID energy on the sign of an external charge could rationalize the reaction acceleration rate in microdroplet experiments[18,19], as the reaction barrier would be decreased in the presence of a positive external charge. Biochemical reactions in certain pH environment is yet another illustration, where a qualitatively correct description requires an accurate treatment of external charges from both electrostatic and electrodynamic points of view.

The analytic approach developed in this work can be straightforwardly incorporated in classical force fields and dispersion-inclusive DFT calculations. All the ingredients needed for evaluating Eq. (2) are readily available in polarizable force fields[38]. The developed approach for FID energy could also be coarse grained and incorporated in continuum solvation models, which are up to now devoid of any coupling between electrostatic and electrodynamic effects. Further extension of our theory is also possible by including higher-order diagrams akin to that shown in Fig. 2i. The FID energy stemming from such higher-order diagrams is asymmetric with respect to the sign of the external charge (unlike Eq. (2)) and thus higher-order effects could become important for modeling the fine details of the field-molecule coupling.

In a more general context, our study indicates the possibility of a noticeable coupling between electrostatic multipole moments with electrodynamic fluctuating moments. This indicates that both static and dynamic intermolecular interactions must be treated on an equal footing, and that the QDO model[23–26] provides an adequate approach that includes proper coupling between electrostatics, polarization, and dispersion for systems in isolation or subject to external fields. As vdW dispersion interactions can also have a direct effect on the electron charge distribution in large molecules and materials[63,64], we emphasize the need to treat classical electrostatics and quantum vdW interactions in a unified and self-consistent manner. Only such advanced methods will ultimately allow to achieve predictive power in atomistic modeling of complex molecular materials.

## Methods

**FID derivation**. Expressions of FID energies are derived building the Rayleigh-Schrödinger perturbation theory (RSPT) on unperturbed QDO eigenfunctions. First three consecutive orders of RSPT are given by the well-known equations:

$$
\begin{aligned}
E^{(1)} &= \langle 0|H'|0\rangle, \\
E^{(2)} &= \sum_{k\neq 0} \frac{|\langle 0|H'|k\rangle|^2}{(E^{(0)}-E^{(k)})}
\end{aligned}
\tag{1}
$$

$$
E^{(3)} = \sum_{k\neq 0,m\neq 0} \frac{\langle 0|H'|m\rangle\langle m|H'|k\rangle\langle k|H'|0\rangle}{(E^{(0)}-E^{(k)})(E^{(0)}-E^{(m)})} - \langle 0|H'|0\rangle \sum_{m\neq 0} \frac{|\langle 0|H'|m\rangle|^2}{(E^{(0)}-E^{(m)})^2},
\tag{2}
$$

where the perturbation includes the interaction between QDOs and influence of the external field $H' = H_A + H_B + H_{AB}$. Interaction of each QDO with external charge in terms of spherical multipole tensor has the form: $H_{A/B} = -\delta \sum_{l=1}^{\infty} \sum_{m=-l}^{m=l} Q_{lm}^{A/B}/\tilde{R}_{A/B}^{l+1}$, where $\tilde{R}_{A/B}$ is a distance between QDO and external charge. Using addition theorem for spherical harmonics one can factorize the Coulomb potential and write the Hamiltonian of the interacting multipoles in the form: $H_{AB} = \sum_{l,l'=0}^{\infty} \sum_{m,m'} Q_{lm}^A T_{lm;l'm'}^{AB}(\mathbf{R}) Q_{l'm'}^B$, where $\mathbf{R}$ is a vector connecting the centers of multipoles, $Q_{lm}^{A/B}$ are complex multipole tensors in a spherical representation and

defined in the global coordinate system. The multipole interaction function takes the form:

$$
\begin{aligned}
T_{l_A m_A l_B m_B}(\mathbf{R}) &= (-1)^{l_A} \sqrt{\frac{(2l_A+2l_B+1)!}{(2l_A)!(2l_B)!}} \\
&\times \begin{pmatrix} l_A & l_B & l_A+l_B \\ m_A & m_B & -(m_A+m_B) \end{pmatrix} I_{l_A+l_B,-(m_A+m_B)}(\mathbf{R}),
\end{aligned}
\tag{3}
$$

where $I_{l,m}(\mathbf{R})$ are normalized irregular spherical harmonics and large brackets represent Wigner $3j$ symbol. FID term arises from the matrix element proportional to $\langle 0|H_{AB}|m\rangle\langle m|H_A|k\rangle\langle k|H_{AB}|0\rangle$ and symmetric one with respect to $A \leftrightarrows B$ substitution. In case of spherically symmetric QDO, matrix elements that appear in the perturbation series can be calculated explicitly.

**Ab initio calculations**. RPA and MP2 calculations[51,52] were carried out with the FHI-aims code[65]. Numerical results are obtained using different exchange-correlation functionals (to assess the role of self-interaction error): PBE[66], Hartree-Fock and hybrid PBE0[67,68] with atom-centered orbital(NAO) basis sets with valence-correlation consistency (VCC), designed to contain explicit sums over unoccupied states[69] within full-potential all-electron approach. The basis set incompleteness error was reduced by the two-point extrapolation scheme applied to the basis set sequence NAO-VCC-4Z and NAO-VCC-5Z[69]:

$$
E(\infty) = \frac{E(n_1)n_1^3 - E(n_2)n_2^3}{n_1^3 - n_2^3}
\tag{4}
$$

where $n_{1,2}$—are indexes of NAO-VCC-nZ basis sets, being equal to 4 and 5 in this work.

**Data availability**. The data that support the findings of this study are available from the corresponding author upon reasonable request.

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

## Acknowledgements

We acknowledge useful discussions with M. Sadhukhan and J. Hermann. We also acknowledge generous funding by the European Research Council (ERC-CoG BeStMo) and Deutsche Forschungsgemeinschaft (DFG-SFB 951 project A10).

## Author contributions

A.T. conceived the physical model to describe FID and the molecular systems to demonstrate FID quantitatively. A.K. derived the FID analytic expressions and carried out all calculations. Both authors discussed the contents and wrote the manuscript.

## Additional information

**Competing interests:** The authors declare no competing interests.

