## [Peer Review File · Nature Communications]

Reviewers' comments:

Reviewer #1 (Remarks to the Author):

This is a fascinating and well-written manuscript developing methods for quantifying the impact of external electric fields on dispersion interactions. The results are striking, with implications for understanding a dizzying array of chemical systems. The authors demonstrate the shockingly large effect of external electric fields on both prototypical small molecular dimers as well as in the context of biological systems.

I see no means of improving this manuscript and look forward to seeing it published!

Reviewer #2 (Remarks to the Author):

The paper by Tkatchenko concerns an interesting subject but proposes marginal incremental novelties with respect to well known approaches. Also the results are quite interesting but not really impressive. On these grounds I think the paper can be published on a specialized journal but does not reach the standards of novelty and generality required for publication on nature communications

Reviewer #3 (Remarks to the Author):

In this paper the authors develop an analytical model and study the effects of an external charge on long-range van der Waals interactions (field-induced dispersion). This is of clear importance for tailoring intermolecular interactions by suitable electric fields, with many possible practical applications, i.e., in material science, biophysics,...

Analytical results are applied to a series of dimers (some of them of biological relevance) validated by comparison with high-level correlated quantum chemistry methods.

The proposed model could be easily incorporated in classical force fields and dispersion-corrected DFT calculations.

Suggestions for further, possible improvements of the scheme are also given.

I find the paper very interesting and useful for a large community of researchers and I judge that is suitable for publication, after the authors have taken the following suggestions into account, which could improve the work presentation.

At page 2, the statement:

"... we project the valence electronic response of a given molecule onto a set of interacting atomic response functions." is rather cryptic and should be clarified.

The following statement, at page 8, should also be better rephrased:

"Provided that vdW dispersion interactions can have a direct effect on the electron charge distribution in large molecules and materials[60, 61] points to the need to develop methods that treat classical electrostatics and quantum vdW interactions in a unified and self-consistent manner."

For the sake of completeness, at page 2, when the authors introduce the

QDO model and state that it yields an accurate description of many-body effects, they should cite the following paper where this is demonstrated: A. Tkatchenko et al., J. Chem. Phys. 138 (2013) 074106.

Eqs. (2) and (3) describe how the effect of external charges decay as a function of the distance from the QDOs. Do the authors expect that even external dipoles could have a significant effect at realistic distances, although with a faster decay? A comment on this would be useful.

Fig.(3) seems to suggest that only the distance between the external charge and a single QDO is relevant, while the text mentions the distance from each QDO. I suggest to slightly modify Fig.(3) in order to avoid any misunderstanding.

At page 5 it is stated that both RPA and analytic interaction energy "are essentially symmetric with respect to the sign of the external charge". But then, why is W_c^{bind} different from zero? A clarification on this point would be useful.

The finding that MP2 binding energies are not symmetric with respect to the sign of the external charge is a non-trivial result which would deserve a comment. Why do the author explain such a behavior, compared with the more symmetric one of RPA?

We are grateful to all referees for taking their time to provide valuable comments and suggestions. We have carefully reviewed the comments and revised the manuscript accordingly. We have also adjusted the manuscript details according to the guidelines of Nature Communications. Our responses are given in a point-by-point manner below.

Reviewer #1:

This is a fascinating and well-written manuscript developing methods for quantifying the impact of external electric fields on dispersion interactions. The results are striking, with implications for understanding a dizzying array of chemical systems. The authors demonstrate the shockingly large effect of external electric fields on both prototypical small molecular dimers as well as in the context of biological systems.

I see no means of improving this manuscript and look forward to seeing it published!

We sincerely thank the reviewer for evaluating our manuscript very positively and highlighting the wide implications of our results for many chemical systems.

Reviewer #2:

The paper by Tkatchenko concerns an interesting subject but proposes marginal incremental novelties with respect to well known approaches. Also the results are quite interesting but not really impressive. On these grounds I think the paper can be published on a specialized journal but does not reach the standards of novelty and generality required for publication on nature communications

Reviewer #2 seems to question the novelty of our work, but he/she did not provide any evidence to support his/her statements. We are not aware of any previously published work (or work under consideration elsewhere) that models the effect of external charges on van der Waals dispersion interactions. Hence, we strongly disagree with reviewer #2 concerning the novelty and impact of our work. This view is also firmly supported by the highly positive reports of reviewers #1 and #3.

Reviewer #3:

In this paper the authors develop an analytical model and study the effects of an external charge on long-range van der Waals interactions (field-induced dispersion). This is of clear importance for tailoring intermolecular interactions by suitable electric fields, with many possible practical applications, i.e., in material science, biophysics,...

Analytical results are applied to a series of dimers (some of them of biological relevance) validated by comparison with high-level correlated quantum chemistry methods.

The proposed model could be easily incorporated in classical force fields and dispersion-corrected DFT calculations.

Suggestions for further, possible improvements of the scheme are

also given.

I find the paper very interesting and useful for a large community of researchers and I judge that is suitable for publication, after the authors have taken the following suggestions into account, which could improve the work presentation.

We thank the referee for his/her very positive evaluation of our manuscript and the constructive suggestions to improve our presentation. We address all these comments below.

At page 2, the statement:

"... we project the valence electronic response of a given molecule onto a set of interacting atomic response functions." is rather cryptic and should be clarified.

We thank the referee for the comment. We rephrase this in the following way:

"we treat the response of valence electrons of a given molecule as that of a set of interacting atomic response functions."

The following statement, at page 8, should also be better rephrased:

"Provided that vdW dispersion interactions can have a direct effect on the electron charge distribution in large molecules and materials[60, 61] points to the need to develop methods that treat classical electrostatics and quantum vdW interactions in a unified and self-consistent manner."

We thank the referee for this comment and rephrase the sentence as follows:

"Since vdW dispersion interactions can also have a direct effect on the electron charge distribution in large molecules and materials [61, 62], we emphasize the need to treat classical electrostatics and quantum vdW interactions in a unified and self-consistent manner."

For the sake of completeness, at page 2, when the authors introduce the QDO model and state that it yields an accurate description of many-body effects, they should cite the following paper where this is demonstrated: A. Tkatchenko et al., J. Chem. Phys. 138 (2013) 074106.

We thank the referee for the suggestion and we added this reference.

Eqs. (2) and (3) describe how the effect of external charges decay as a function of the distance from the QDOs. Do the authors expect that even external dipoles could have a significant effect at realistic distances, although with a faster decay ? A comment on this would be useful.

We thank the referee for raising this question. Similar effect is always present in case of non-uniform electric fields. Therefore, for a finite-size dipole, higher order of perturbation theory would result in a similar term with decay $\sim R^{-9}$. In order to clarify this issue we add to the paragraph after Eq. 2:

“Similar effect is expected in case of other non-uniform electric fields, for example that induced by a finite-size dipole. FID would scale as \tilde{R}^{-9} with a distance to the dipole, making dispersion induced by a point charge a leading order effect. “

Fig.(3) seems to suggest that only the distance between the external charge and a single QDO is relevant, while the text mentions the distance from each QDO. I suggest to slightly modify Fig.(3) in order to avoid any misunderstanding.

We thank the referee for raising this issue. To avoid possible confusion, we updated Figure 3 by adding a dashed line connecting the point charge to the center of mass of both molecules.

At page 5 it is stated that both RPA and analytic interaction energy "are essentially symmetric with respect to the sign of the external charge". But then, why is W_c^{bind} different from zero? A clarification on this point would be useful.

We thank referee for this comment, indeed the FID energies are antisymmetric, as follows from Eq.(2) and Eq.(3). Binding energies are governed by the following relation: $E_c^{\text{bind}}(-\delta) = -E_c^{\text{bind}}(\delta)$. Therefore, analytics suggest that W_c^{bind} is just $2x E_c^{\text{bind}}(-\delta)$. In order to avoid confusion we change this sentence to: “are essentially antisymmetric with respect to the sign of the external charge”.

The finding that MP2 binding energies are not symmetric with respect to the sign of the external charge is a non-trivial result which would deserve a comment. Why do the author explain such a behavior, compared with the more symmetric one of RPA?

We agree that a clarification is required. The non-symmetric behavior of MP2 correlation energy with respect to the sign of external charge is mainly governed by the response of Hartree-Fock wavefunction. When the external charge is close to the molecule it gives rise to non-linear response to electric fields (hyperpolarization) that result in a non-symmetric change in the correlation energy. RPA, in contrast to MP2, is more robust at short and large distances, due to its ability to include all orders of correlation contributions in the linear response. To clarify this, we added a sentence in the “FID interaction between small molecular dimers” section at the end of second paragraph:

“However, MP2 binding energies in this case arise from hyperpolarization effects (not linear polarization as in RPA) and are not antisymmetric with respect to $\delta \rightarrow -\delta$, having a smaller magnitude compared to the reference RPA calculations (also see Table I). This is particularly significant for practically relevant distances, when the external charge is close to the molecule.”

REVIEWERS' COMMENTS:

Reviewer #3:

In my opinion, the suggestions/criticisms raised by all the referees have been satisfactorily addressed by the authors in their response letter. Therefore I judge that the revised version of the paper is suitable for publication.